# DER-Solomon: A Large Number of CVRPTW Instances Generated Based on the Solomon Benchmark Distribution

## Abstract

The Solomon benchmark is a well-known resource for researching Capacitated Vehicle Routing Problem with Time Windows (CVRPTW), and has been used by many traditional methods. However, the limited scale of the Solomon benchmark poses challenges to effective utilization by learning-based approaches. To address this, we propose an expanded version with a large set of new instances, called DER-Solomon benchmark, which follows a similar distribution as the Solomon benchmark. First, we analyze the Solomon benchmark and use backward derivation to establish an approximate distribution, from which the DER-Solomon is generated, thereby significantly expanding the size of the benchmark. Next, we validate the distribution consistency between the DER-Solomon benchmark and the original Solomon benchmark using traditional algorithms. We then demonstrate the superiority and reliability of DER-Solomon compared to other similar Solomon-like datasets using state-of-the-art Deep Reinforcement Learning (DRL) algorithms. Finally, we train multiple DRL algorithms using the DER-Solomon benchmark and compare them with the traditional algorithms. The results show that the DRL algorithms trained on the DER-Solomon benchmark can achieve the same level of solution quality as the traditional algorithms on the Solomon benchmark while reducing the computational time by over 1000 times on CVRPTW. All the results demonstrate that the DER-Solomon benchmark is sufficiently excellent, serving as an extension of the Solomon benchmark, which offers valuable tools and resources for further research and solutions to the CVRPTW problem.

## 1 Introduction

The Capacitated vehicle routing problem (CVRP) is a classical combinatorial optimization problem, which aims to optimize the routes for a fleet of vehicles with capacity constraints to serve a set of customers with demands (Dantzig et al., 1954; Li et al., 2021). Compared with the assumption of customers is no service time requirement, the settings of customers with different time window constraints are more in line with the real-world practice, which leads to the CVRP with Time Windows (CVRPTW) (Solomon, 1987; Desaulniers et al., 2016). The objective of the problem is to devise a set of routes that efficiently serve each customer within their respective time windows and capacity constraints for all vehicles, while minimizing the total traveling distance or the overall transportation cost (Bräysy & Gendreau, 2005; Schneider et al., 2014; Wang et al., 2021b). The CVRPTW is a crucial aspect of logistics and transportation, attracting extensive research efforts to find effective solutions. Numerous methods have been proposed to tackle this problem from conventional methods to DRL-based methods.

The Solomon benchmark is a set of instances proposed by Solomon (1987). It is based on real-life data and considers practical constraints, providing standardized CVRPTW instances. The Solomon benchmark is complex enough to test all aspects of algorithms, and has been cited as a public test dataset for a variety of algorithms in the CVRPTW field for years, ranging from heuristic algorithms to evolutionary algorithms, from single-objective problems to multi-objective problems (Kuo et al., 2022; Chen et al., 2021; Luo et al., 2018; 2016).

Exact algorithms obtain or approximate optimal solutions by exhaustively enumerating all possibilities (Yang, 2023; Desaulniers et al., 2014). Researchers have primarily focused on exact algorithms for the CVRPTW using the Solomon benchmark as experimental data. These studies have produced highly favorable outcomes. Various exact algorithms have been extensively researched, including branch and bound, branch and price, and dynamic programming. Macedo proposed an iterative exact algorithm for this problem, which depends on a pseudo-polynomial network flow model in which nodes represent time points, and arcs represent feasible vehicle routes (Macedo et al., 2011). Costa et al. emphasized the major methodological and modeling contributions over the years on branch and price algorithms, whether general or specific to VRP variants (Costa et al., 2019). Baldacci et al. surveyed methods based on linear and integer programming and compared the performance and complexity of branch and price methods and branch and bound methods (Baldacci et al., 2012).

Heuristic algorithms are methods that continuously optimize the current solutions to search for better solutions (Bräysy & Gendreau, 2005). Various heuristic algorithms for the CVRPTW problem have been extensively researched and applied, including genetic algorithms (Berger & Barkaoui, 2004; Putri et al., 2021), simulated annealing algorithms (Czech & Czarnas, 2002), tabu search algorithms (Cordeau & Maischberger, 2012), ant colony algorithms (Razavi & Eshlaghy, 2015), etc. The comparison between these algorithms usually uses the Solomon benchmark as test data. Evolutionary algorithms, which simulate the natural evolution process to search for optimal solutions in the solution space (González et al., 2018; Cybula et al., 2022), have also been widely studied and applied to the CVRPTW problem, such as genetic algorithms (Liu & Jiang, 2019) and differential evolution algorithms (Pitakaso et al., 2020). Marrouche et al. studied a population-based metaheuristic algorithm, specifically Strength Pareto Evolutionary Algorithm 2 (SPEA2) with local search features (Marrouche & Harmanani, 2021).

Due to the strong NP-completeness of the CVRPTW problem, traditional algorithms like exact and heuristic algorithms have limitations such as high computational complexity, long execution time, and poor portability (Zhang et al., 2022; Wang et al., 2021a). Data-driven approaches such as Deep Reinforcement Learning (DRL) algorithms have become a breakthrough in overcoming the limitations of traditional algorithms by improving the efficiency of solving combinatorial optimization problems through learned experiences and strategies (Bello et al., 2017; Kool et al., 2019; Zong et al., 2022). Still, they require a large number of instances during the training process (Lin et al., 2021; Zhang et al., 2020; Liang et al., 2023). However, the Solomon benchmark only consists of 56 instances, which is a limited number for DRL algorithms to solve the CVRPTW problem. Currently, the majority of DRL algorithms for CVRPTW are trained using generated datasets similar to the Solomon benchmark (referred to as the "Solomon-like benchmark" in this paper). However, these generated datasets lack theoretical backward derivation and rely solely on estimated Gaussian distributions to generate instances without undergoing similarity verification. Consequently, a significant disparity exists when comparing these datasets with the original Solomon benchmark. As a result, DRL algorithms are often evaluated on partial customer points or datasets resembling the Solomon benchmark (Xu et al., 2021; Zhang et al., 2020), without directly on the original benchmark itself. Hence, an urgent requirement is to expand the number of instances matching the original Solomon benchmark distribution.

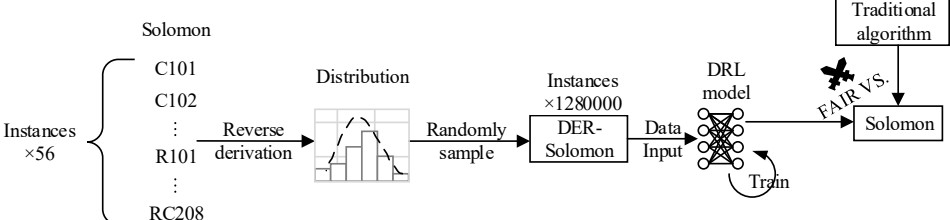

Figure 1: Schematic diagram of the DER-Solomon benchmark generation process. The goal is for DRL methods, trained on DER-Solomon, to compete fairly with traditional methods on the Solomon benchmark.

This paper aims to generate an improved Solomon benchmark with a large number of instances for CVRPTW, called the derived-Solomon (DER-Solomon) benchmark. To this end, we analyze the

characteristics of the 56 instances of the Solomon benchmark in detail, and backward derive the distribution of the time windows. The process of backward deriving is shown as in Figure 1. By comparing the testing results of traditional algorithms on the original Solomon benchmark and the DER-Solomon benchmark, we demonstrate that the data generated by our backward deriving method has a similar distribution to the Solomon benchmark. Furthermore, we use the DER-Solomon benchmark and the existing Solomon-like benchmark, respectively to train MARDAM(Bono et al., 2020), a state-of-the-art DRL algorithm for CVRPTW. Subsequently, we evaluate the outcomes of the trained models using the standard Solomon benchmark. The experimental results show that the dataset of DER-Solomon generated by our method is closer to the Solomon benchmark compared to other similar datasets. Finally, We successfully apply various advanced DRL methods to solve the CVRPTW and train them with the DER-Solomon benchmark to achieve a fair comparison of the performance between the DRL and traditional algorithms on the Solomon benchmark. The contributions of this paper are as follows:

- By backward deriving, we establish the distribution model of the 56 instances in the Solomon benchmark and generate the DER-Solomon benchmark, thus expanding the number of instances in the benchmark.

- We verify the distribution consistency between the DER-Solomon benchmark and the Solomon benchmark using traditional algorithms. Moreover, we demonstrate the superiority and reliability of the DER-Solomon benchmark in DRL algorithms, showing that it is closer to the Solomon benchmark than other similar Solomon-like benchmark.

- We propose a method for scaling up the Solomon benchmark through backward deriving. This method serves as a valuable reference for enlarging the scale of other combinatorial optimization datasets, facilitating their further advancement.

- The DER-Solomon benchmark complements the number of instances in the Solomon benchmark, facilitating fair comparisons between DRL-based CVRPTW solving algorithms and traditional algorithms on the standard test instances of the Solomon benchmark. It also facilitates comparisons of various DRL-based CVRPTW solving algorithms on DER-Solomon instances with the same complexity and sufficient quantity as Solomon benchmark, promoting the development and improvement of related algorithms.

## 2 BACKWARD DERIVATION MODEL

For a data set $X = \{x_1, x_2, \cdots, x_n\}$, its frequency histogram is shown in Figure 2(a). To infer the appropriate distribution function that matches the Solomon benchmark datasets, we begin by assuming multiple forms of distribution functions. For each distribution function form, parameter estimation is performed. The goodness of fit for each distribution function is evaluated by calculating the Residual Sum of Squares (RSS). The distribution function with the smallest RSS is selected as the best-fitting distribution, as evidenced by the fitting curve depicted in Figure 2(b). The entire process is visually represented in Figure 3.

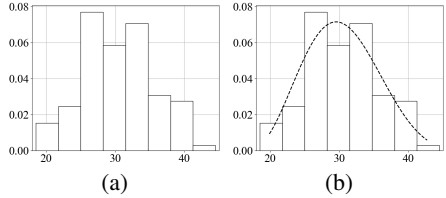

Figure 2: Frequency histogram of data X (a) and its fitting distribution curve (b)

### 2.1 MODEL DERIVATION

The popular distribution function models include Normal, Exponential, Pareto, Weibull, $t$, General extreme (GE), Gamma, log Normal, Beta, Uniform, and Log Gamma. Using these models to infer the distribution of variables allows for coverage of the true distribution of variables in most cases.

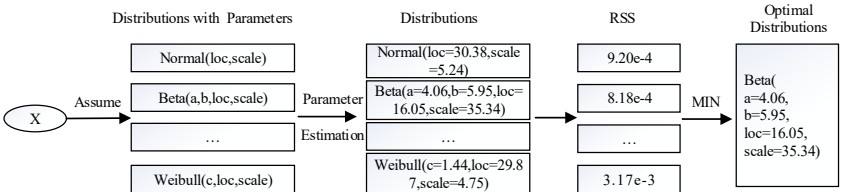

Figure 3: Backward derivation process of data set X's distribution function. By using various popular distribution functions in statistical techniques, we can derive the data X to approximate its true distribution. This is particularly suitable for scenarios with a small amount of data.

In the backward derivation of Solomon benchmark distribution, after experimental testing, three distribution function forms were selected with the best fit: Weibull, GE, and Beta. Their distribution function forms are given by formulae (1), (2), and (3). To facilitate parameter fitting, let $y = (x - loc)/scale$, where $loc$ and $scale$ are the translation and scaling parameters of the independent variable.

$$\text{Weibull}(x; c) = \frac{c}{2}|y|^{c-1}e^{-|y|^c}, y \in R, c > 0. \tag{1}$$

$$\text{GE}(x; c) = \begin{cases} e^{-e^{-y}}e^{-y} & , c = 0 \\ e^{-(1-cy)^{\frac{1}{c}}}(1 - cy)^{\frac{1}{c}-1}, c \neq 0, \end{cases} \tag{2}$$

where $-\infty < y \leq \frac{1}{c}$ if $c < 0$, and $\frac{1}{c} \leq y < \infty$ if $c > 0$.

$$\text{Beta}(x; a, \ b) = \frac{\Gamma(a + b)y^{a-1}(1 - y)^{b-1}}{\Gamma(a)\Gamma(b)}, \tag{3}$$

where

$$\Gamma(x) = \int_0^{+\infty} t^{y-1}e^{-t}dt, 0 \leq y \leq 1, a > 0, b > 0. \tag{4}$$

## 2.2 MODEL PARAMETERS ESTIMATION

In statistics, maximum likelihood estimation (MLE) is a method of estimating the parameters of an assumed probability distribution, given some observed data (Wikipedia contributors, 2023). For a data set $X = \{x_1, x_2, \cdots, x_n\}$, whose distribution function form is assumed to be $f(x; \Theta)$ with parameter $\Theta$, its likelihood function is built as

$$L(\Theta) = \ln\left\{\prod_{i=1}^n f(x_i; \Theta)\right\}^{\frac{1}{n}} = \frac{1}{n}\sum_{i=1}^n \ln(f(x_i; \Theta)). \tag{5}$$

And then make it maximum to solve the $\Theta$. Intuitively, this selects the parameter values that make the observed data most probable.

To solve it, we firstly use MINIPACK (Moré et al., 1980), which is a FORTRAN program used to solve nonlinear least squares problems. And the initial solution in MINIPACK of $\Theta$ is set to 1 with the exception of $loc$ and $scale$ in it, which are set to the mean and variance, respectively, of $X$. The solution obtained from MINPACK is then used as the initial values for further optimization using the Nelder-Mead method (Gao & Han, 2012). The pseudocode for the Nelder-Mead algorithm is shown in appendix.

## 2.3 RSS TEST

A goodness-of-fit test performed on a fitted distribution function can evaluate it. The Residual Sum of Squares (RSS) describes the deviation between the predicted data from the distribution function

and the real data. It quantity the gap between the fitted distribution function and the real distribution function, with a smaller RSS indicating a closer fit between the model and the data. RSS is calculated as

$$RSS = \sum_{i=1}^{n} \left( y_i - f\left( x_i \right) \right)^2 \tag{6}$$

where $y_i$ represents the actual data values, $x_i$ is the independent variable, and $f\left( x_i \right)$ is the predicted value from the model. The fitting performances of different distribution functions to the data are evaluated by performing a goodness-of-fit test using RSS. Subsequently, by comparing and ranking the RSS values, the distribution function with the smallest RSS is chosen as the best theoretical distribution for the data.

## 3 DERIVATION OF SOLOMON BENCHMARK DISTRIBUTION

This section explores the analysis of the Solomon benchmark and presents the derivation process and results of the Solomon benchmark in conjunction with the mathematical model discussed in the previous section.

### 3.1 SOLOMON BENCHMARK

The Solomon benchmark, proposed by Solomon (1987), serves as a public test dataset for the CVRPTW algorithm. It consists of six series: C1, C2, R1, R2, RC1, RC2, each containing 8 to 12 different instances. Each instance includes 101 data points, consisting of customer point numbers, x and y coordinates, demand, time window start time, time window end time, and service duration. The first data point (NO. 0) corresponds to the depot node. The maximum capacity Q of vehicles is the same within the same series, and the corresponding Q values for each series are: 200, 700, 200, 1000, 200, and 1000.

In different instances within the same series, such as C101 and C102, the customer points with identical numbers have the same x and y coordinates, demand, and service duration. The only difference is their time window. Therefore, to generate a large number of data similar to the Solomon benchmark distribution for DRL algorithm training, DRL algorithms can be trained separately in each series so as to keep the x and y coordinates, demand, and service duration the same as the corresponding series of Solomon benchmark, only randomly generating the time windows according to the Solomon benchmark distribution.

### 3.2 TIME WINDOW DENSITY

In the Solomon benchmark, the time window density may vary for different instances. Time window density indicates the percentage that the number of customer points subject to time window constraints among the total number of customer points. There are four levels of time window density: 25%, 50%, 75%, and 100%. Analysis reveals that instances with densities of 25%, 50%, and 75% are low-density versions of an instance in the same series with a density of 100%. For example, in the C1 series, the time windows for C102, C103, and C104 are versions with densities of 75%, 50%, and 25%, respectively, compared to C101. This means that they use the same time windows as C101, but only a portion (75%, 50% and 25% respective) of customer points of them need to satisfy the time window constraints.

### 3.3 TIME WINDOW WIDTH

The generation of time windows depends on two factors: the center values and the width of the time windows (Solomon, 1987), rather than the start time and the end time. The center values represents the intermediate time between the start time and the end time. The width represents the duration from the start time to the end time. In the Solomon benchmark, the center values of time windows for R and RC series are uniformly and randomly generated within range $[\text{ENTER}_0 + \text{distance}(0, i),$ $\text{LEAVE}_0 - \text{distance}(0, i) - \text{SERVICE}_i]$, where $\text{ENTER}_0$ and $\text{LEAVE}_0$ are the time window of the depot point, distance $(0, i)$ is the travel time between customer point $i$ and the depot, and $\text{SERVICE}_i$ represents the time required to serve customer point $i$. As for the C series, the center values of time

windows are obtained by solving the CVRP problem without time window constraints using the 3-opt route method, which provides the arrival times for each customer point as the center values. In other words, the center values of time windows for the Solomon benchmark instances are fixed (series C) or have a known distribution function (R and RC). Therefore, the Solomon benchmark distribution's derivation only involves the distribution of time window width. For ease of use in code, this paper deduces the distribution of half-widths of time windows.

### 3.4 BACKWARD DERIVATION

To backward derive a distribution function of an instance, taking C101 as an example, following the process illustrated in Figure 3, we first assume multiple distribution forms for the width, such as Normal, Exponential, Pareto, Weibull, t, GE, Gamma, log Normal, Beta, Uniform, and log Gamma. Then, we estimate parameters for all distribution forms. Taking the Beta function shown in formula (3) as an example, we construct the maximization objective function for the maximum likelihood estimation according to the formula (5). Let $\Theta = [a, b, loc, scale]^{\mathrm{T}}$, and we have

$$L(\Theta) = \frac{1}{100} \sum_{i=1}^{100} \ln(\mathrm{Beta}(x = x_i; \Theta)), \tag{7}$$

where the $x_i$ represents the half-width of the time window of the $i^{\mathrm{th}}$ customer node of C101. We then use MINIPACK to minimize $-L(\Theta)$ preliminarily and obtain $\Theta = [3.48, 5.33, 17.13, 33.55]^{\mathrm{T}}$. Based on this, we set the initial values of Nelder-Mead to $1 + \sigma$ times the $\Theta$ on each parameter, i.e., $\Theta_5 = \Theta$, and $\Theta_1, \Theta_2, \Theta_3, \Theta_4$ are $[3.48 \times 1.05, 5.33, 17.13, 33.55]^{\mathrm{T}}, \cdots, [3.48, 5.33, 17.13, 33.55 \times 1.05]^{\mathrm{T}}$. After 600 iterations, we obtain $\Theta = [4.06, 5.95, 16.05, 35.34]^{\mathrm{T}}$, indicating that the distribution function is $\mathrm{Beta}(a = 4.06, b = 5.95, loc = 16.05, scale = 35.34)$, corresponding to the distribution curve in Figure 2(b).

To test the fitting distribution functions, we conduct the RSS goodness-of-fit test. In Figure 2(b), the frequencies of histogram bin centers are $\{0.0154, 0.0246, \cdots, 0.00308\}$. The predicted values of the distribution function $\mathrm{beta}(a = 4.06, b = 5.95, loc = 16.05, scale = 35.34)$ at the bin centers are $\{0.0108, 0.0378, \cdots, 0.00552\}$. The RSS is calculated as the sum of the squared differences between the frequencies and the predicted values, giving $RSS = (0.0154 - 0.0108)^2 + (0.0246 - 0.0378)^2 + \cdots + (0.00308 - 0.00552)^2 = 8.18\mathrm{e} - 4$.

Similarly, the RSS values for the remaining commonly used distribution forms, i.e., Normal, Exponential, Pareto, Weibull, t, GE, Gamma, log Normal, Uniform, and log Gamma, are estimated as follows: $9.20\mathrm{e} - 4, 8.73\mathrm{e} - 3, 8.73\mathrm{e} - 3, 3.16\mathrm{e} - 3, 9.20\mathrm{e} - 4, 8.24\mathrm{e} - 4, 8.56\mathrm{e} - 4, 8.61\mathrm{e} - 4, 5.07\mathrm{e} - 3, 9.30\mathrm{e} - 4$. Among all distribution forms, the Beta distribution has the smallest RSS value of $8.18\mathrm{e} - 4$, indicating that it is the best theoretical distribution, namely Beta $(a = 4.06, b = 5.95, loc = 16.05, scale = 35.34)$ distribution.

The detailed derivation results for 56 instances are shown in Table 3 of appendix, and the distribution functions used for generating data for each series are listed in Table 4 of appendix.

## 4 EXPERIMENTS

To verify the effectiveness of the proposed DER-Solomon benchmark in comparison to the Solomon benchmark, we conduct four groups of experiments. In the first group of experiments, we use traditional algorithms to solve the DER-Solomon benchmark and Solomon benchmark to verify the consistency between the two datasets. In the second group of experiments, we train the state-of-the-art DRL algorithm designed for CVRPTW using the DER-Solomon and Solomon-like benchmarks, respectively, and test it on the Solomon benchmark to demonstrate the superiority of the DER-Solomon benchmark. In the third group of experiments, we apply various advanced DRL algorithms originally solving CVRP to solve CVRPTW and train them with the DER-Solomon benchmark, then compare them with three traditional algorithms on the Solomon benchmark to show their performance on such fair comparison. In the fourth group of experiments, we show the time cost of those two sets of algorithms on a large number of CVRPTW instances to demonstrate the advantages of DRL algorithms in terms of computing efficiency.

### 4.1 SIMILARITY BETWEEN DER-SOLOMON AND SOLOMON BENCHMARK WITH TRADITIONAL ALGORITHMS

To verify the DER-Solomon benchmark and the Solomon benchmark have a similar effect on algorithms, we used the following three traditional algorithms to test these two datasets and compared the mean and standard deviation of the test results. These three traditional algorithms are: LKH3 (Helsgaun, 2017), a famous heuristic solver that achieves state-of-the-art performance on various routing problems; OR-Tools (Perron & Furnon, 2019), a mature and widely used routing problem solver based on meta-heuristics; GA (Ombuki et al., 2006), an advanced routing problem solver based on genetic algorithm.

Table 1 shows the mean and standard deviation of the test results for GA, LKH, and OR-Tools on the Solomon benchmark and DER-Solomon benchmark, respectively, and the mean and standard deviation Gap between the test results on the two datasets. It can be observed that the Gap in mean does not exceed 5%, and the Gap in standard deviation does not exceed 3%. The experimental results indicate that DER-Solomon benchmark and Solomon benchmark show significant consistency in terms of the mean and standard deviation of solutions achieved by multiple algorithms, demonstrating that DER-Solomon benchmark can serve as an extension of the Solomon benchmark for training algorithms based on both traditional and DRL methods. The LKH and OR-Tools test codes are released [1], as well as GA [2] (Please note that underscores may change to other symbols when pasted into the browser's address bar).

Table 1: Performance of traditional algorithms on DER-Solomon and Solomon benchmark.

| | | LKH | | | OR-Tools | | | GA | | |
| --- | --- | --- | --- | --- | --- | --- | --- | --- | --- | --- |
| | | Solomon | DER-Solomon | Gap | Solomon | DER-Solomon | Gap | Solomon | DER-Solomon | Gap |
| mean | C1 | 827.3 | 824.8 | 0.30 % | 919.5 | 920.4 | 0.10 % | 833.7 | 836.0 | 0.27 % |
| | C2 | 590.0 | 588.4 | 0.26 % | 631.6 | 616.4 | 2.40 % | 591.5 | 588.6 | 0.50 % |
| | R1 | 1187.2 | 1131.4 | **4.70** % | 1235.9 | 1199.6 | 2.94 % | 1194.9 | 1159 | 3.01 % |
| | R2 | 882.0 | 869.6 | 1.41 % | 963.0 | 965.1 | 0.23 % | 903.1 | 889.4 | 1.52 % |
| | RC1 | 1352.1 | 1292.0 | 4.45 % | 1439.5 | 1384.1 | **3.84** % | 1373.5 | 1310.9 | **4.56** % |
| | RC2 | 1010.7 | 1002.0 | 0.86 % | 1124.6 | 1108.1 | 1.46 % | 1018.5 | 1022.7 | 0.41 % |
| std | C1 | 1.44 | 0.13 | 0.16 % | 88.41 | 65.44 | **2.50** % | 13.00 | 14.18 | 0.14 % |
| | C2 | 1.57 | 0.24 | 0.23 % | 36.32 | 35.30 | 0.16 % | 11.37 | 4.52 | 1.16 % |
| | R1 | 204.29 | 174.09 | 2.54 % | 168.94 | 143.79 | 2.03 % | 208.68 | 187.14 | **1.80** % |
| | R2 | 129.44 | 111.83 | 2.00 % | 142.18 | 119.70 | 2.33 % | 118.56 | 111.69 | 0.76 % |
| | RC1 | 191.03 | 155.24 | **2.65** % | 128.73 | 128.08 | 0.04 % | 173.36 | 151.89 | 1.56 % |
| | RC2 | 172.39 | 148.12 | 2.40 % | 174.13 | 163.59 | 0.94 % | 158.58 | 149.27 | 0.91 % |

The "Gap" is calculate as $\frac{abs(\text{std or mean on Solomon} - \text{std or mean on DER-Solomon})}{\text{mean on Solomon}}$, which represent the difference between Solomon and DER-Solomon. This is a deeper representation of the difference in data using the performance results of various algorithms, rather than simply comparing through the surface features of the data. The **bold** text indicates where DER-Solomon and Solomon show the greatest Gap.

### 4.2 IMPROVEMENT OF DRL ALGORITHMS USING DER-SOLOMON BENCHMARK

In the experiment, we used the open-source MARDAM (Bono et al., 2020) algorithm for CVRPTW to demonstrate the effectiveness of DER-Solomon benchmark. We compare the performance on the Solomon benchmark of the algorithm training with the original data and training with DER-Solomon benchmark, respectively, as shown in Figure 4. The 56 instances on the x-axis correspond to the c1, c2, r1, r2, rc1, and rc2 series' instances from left to right. The MARDAM code is released [1].

As we can seen from Figure 4, the performance of MARDAM trained with DER-Solomon benchmark is significantly better than the one trained with the original training data. The DRL algorithm outperforms the optimal values in some Solomon benchmark instances, which is due to the original

---

[1]https://anonymous.4open.science/r/DER-Solomon_MARDAM-5E47
[2]https://anonymous.4open.science/r/DER-Solomon_GA-62F2

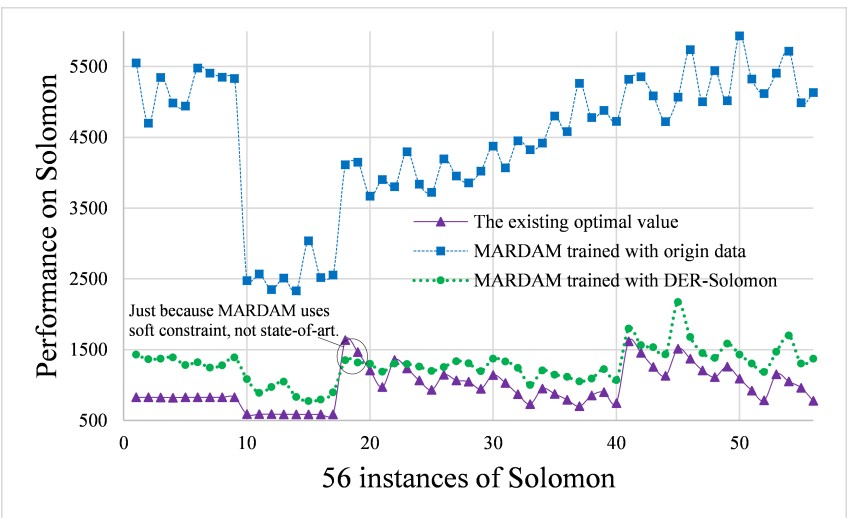

Figure 4: Effect of training with different data. MARDAM trained with DER-Solomon is much better than trained with the original data.

MARDAM model using soft time window constraints. In hard time window constraint problems, the arrival time of a customer point cannot exceed the latest allowable time $\text{ARRIVE}_i < \text{LEAVE}_i$. However, in soft time window problems, there is no such restriction. Instead, soft time window problems involve penalty values $\alpha(\text{LEAVE}_i - \text{ARRIVE}_i)$, where cost = DISTANCE + $\sum_{i \in [1,101]} \alpha(\text{LEAVE}_i - \text{ARRIVE}_i)$, and $\alpha$ is the penalty coefficient, and DISTANCE is the total path length. From this, we can see that DER-Solomon benchmark outperforms other similar Solomon-like benchmark.

### 4.3 Comparison Between DRL and Traditional Algorithms

To enable a fair comparison between DRL algorithms and traditional algorithms on the public test instances of the Solomon benchmark, we trained AM (Kool et al., 2019), MDAM (Xin et al., 2021), and POMO (Kwon et al., 2020) using DER-Solomon benchmark and compared them to traditional algorithms GA, LKH, and OR-Tools on the Solomon benchmark.

When it comes to CVRP problems, the DRL algorithms of AM, MDAM, and POMO are widely acknowledged to be effective. Even though their creators did not offer models specifically for solving CVRPTW, we have adapted their models for CVRP problems to fit the requirements of CVRPTW, and they continue to display strong performance.

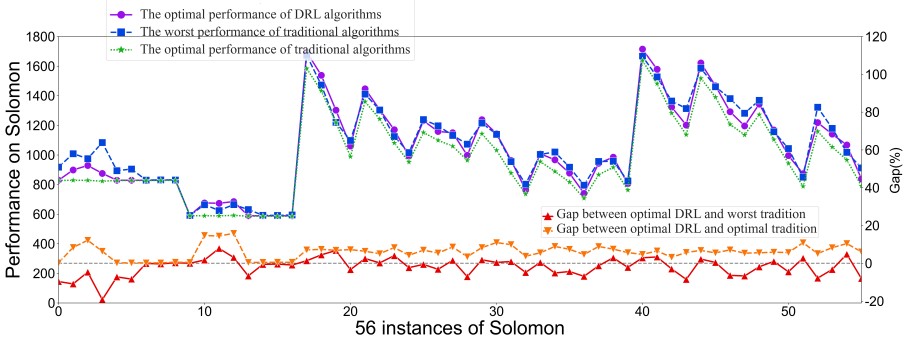

Figure 5: Performance comparison of algorithms on the Solomon benchmark.

Figure 5 extracts the optimal values of the three DRL algorithms, and the worst values and the optimal values of the three traditional algorithms (curves with square, circle, and triangle marks,

respectively) on various instances of the Solomon benchmark. And the detailed values of these algorithms are shown in Table 5 of appendix. In Figure 5, the curves "Gap to max" represents the gap between the optimal value of DRL and the worst value of the traditional algorithm. In contrast, the curves "Gap to min" represents the gap between that of DRL and the optimal value of the traditional algorithm. As seen from the figure, in more than half of the Solomon benchmark instances, DRL algorithms outperformed the worst values of traditional algorithms, and the gaps are within 10% even compared with the optimal values of traditional algorithms, demonstrating that DRL algorithms can achieve a similar level of solution quality as traditional algorithms on the Solomon benchmark. This suggests that training DRL algorithms using DER-Solomon benchmark generated based on the Solomon distribution backward derivation can lead to a fair comparison between DRL algorithms and traditional algorithms on the public test instances of the Solomon benchmark. The code of AM, MDAM, and POMO for solving CVRPTW are released[3].

## 4.4 COMPARISON OF TIME COST

Apart from the competitive solution quality, the computational speed of DRL algorithms is much higher than traditional algorithms. We solved the same 1024 DER-Solomon instances using DRL algorithms (AM, MDAM, POMO) and traditional algorithms (LKH, OR-Tools, GA), and the time taken for each algorithm to solve the instances are shown in Table 2. The experimental results indicate that the computational time of DRL algorithms is significantly lower than that of traditional algorithms, with efficiency improvements exceeding 1000 times. The shorter computational time demonstrates the tremendous potential of DRL algorithms for large-scale CVRPTW planning. Thus, it is worthwhile to explore DRL algorithms further. And facilitating a fair comparison between DRL algorithms and traditional algorithms by training DRL using DER-Solomon is an important part of this.

Table 2: Time taken for algorithms to solve 1024 instances.

|  |  | C1 | C2 | R1 | R2 | RC1 | RC2 |
|---|---|---|---|---|---|---|---|
| Tradition | **LKH** | 25m36s | 47m36s | 46m41s | 1h47m6s | 2h15m40s | 1h44m15s |
|  | OR-Tools | 2h44m36s | 4h1m48s | 3h43m2s | 4h16m12s | 3h4m47s | 4h16m27s |
|  | GA | 4h59m46s | 1h42m29s | 1h9m54s | 5h11m34s | 55m15s | 4h38m52s |
| DRL | **AM** | 1.3s | 1.2s | 1.3s | 1.2s | 1.2s | 1.6s |
|  | MDAM | 13.3s | 11.8s | 12.8s | 12.3s | 12.3s | 11.6s |
|  | POMO | 4.4s | 4.6s | 4.4s | 4.3s | 4.6s | 4.4s |

The **bold** text indicates that LKH and AM are the fastest algorithms in the traditional and DRL algorithms, respectively.

## 5 CONCLUSION

In this study, we use the backward derivation method to fit the distribution of Solomon benchmark and generate DER-Solomon benchmark with a sufficient number of instances and similar complexity to Solomon benchmark. These instances are suitable for training and testing traditional algorithms and DRL algorithms. The DER-Solomon benchmark has demonstrated superior performance compared to other similar Solomon-like benchmarks. The DRL algorithms trained on DER-Solomon benchmark can achieve solutions of equal quality on par with traditional algorithms on the Solomon benchmark, while improving solving time over 1000 times in CVRPTW with a large number of instances compared to traditional algorithms.

The proposed backward derivation method can provide references for generating a large number of simulation instances for other CVRP-related tasks with a limited number of actual instances. DER-Solomon expands the number of Solomon benchmark instances, enabling fair comparisons between DRL algorithms and traditional algorithms.

---

[3]https://anonymous.4open.science/r/DER-Solomon_DRLs-8AA1/

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

# A APPENDIX

## A.1 NELDER-MEAD ALGORITHM

In the Nelder-Mead algorithm1, parameter is set as follows: $\{\sigma, T, \alpha, \beta, \gamma, \delta\} = \{0.05, 600, 1, 2, 0.5, 0.5\}$. $\Theta$ is a column vector with a number of rows equal to the variable $n$ in the algorithm and the number of parameters in the distribution function. For example, in the case of the beta distribution,$n = 4$, and $\Theta = [a, b, loc, scale]^{\mathrm{T}}$.

When Initialization, $\mathbf{E}_n$ is a unit matrix. For $\Theta_0 = [3.48, 5.33, 17.13, 33.55]^{\mathrm{T}}$, $\Theta_1, \Theta_2, \Theta_3, \Theta_4 = [3.48 \times 1.05, \cdots]^{\mathrm{T}}, \cdots, [\cdots, 33.55 \times 1.05]^{\mathrm{T}}$.

## A.2 THE DERIVATION RESULTS OF SOLOMON BENCHMARK DISTRIBUTION

The distribution functions of the 56 instances in the Solomon benchmark are listed in Table 3. In cases with multiple distribution functions, it indicates that the half-widths of time windows for the 100 customer points may follow different distributions.

## A.3 DISTRIBUTION FOR THE DER-SOLOMON GENERATION

The distribution used to generate DER-Solomon is slightly different in form from that of the 56 instances of the Solomon benchmark due to repetition, density variations, etc., and the specific distribution used is shown in Table 4. A total of 1,280,000 instances were generated for training purposes, taking the C1 series as an example, it contains 6 groups of instances with different distributions: Beta(4.06,...), Beta(3.66,...), Gamma(1.52,...), Beta(3.73,...), Constant 90 and Constant 180. Among them, the number of instances with time window half-widths distribution of

---

**Algorithm 1** Nelder-Mead

---

**Input**: $\Theta_0 \leftarrow \text{MINIPACK}(-L(\Theta))$
**Parameter**: $\sigma, T, \alpha, \beta, \gamma, \delta = 0.05, 600, 1, 2, 0.5, 0.5$
**Output**: $\Theta$
    Initialize $\Theta_1, \Theta_2, \cdots, \Theta_n \leftarrow (1 + \sigma)\mathbf{E}_n\Theta_0$.
    $\Theta_{n+1} \leftarrow \Theta_0$.
    Let function $L \leftarrow -L$
    Sort all $\Theta_i$ s.t. $L(\Theta_1) < L(\Theta_2) < \cdots < L(\Theta_{n+1})$.
    **for** $step \leftarrow 1, 2, \cdots, T$ **do**
        $\bar{\Theta} \leftarrow (\Theta_1 + \Theta_2 + \cdots + \Theta_n)/n$
        $\Theta_r \leftarrow \bar{\Theta} + \alpha(\bar{\Theta} - \Theta_{n+1})$
        **if** $L(\Theta_1) \leq L(\Theta_r) < L(\Theta_n)$ **then**
            $\Theta_{n+1} \leftarrow \Theta_r$
        **else if** $L(\Theta_r) < L(\Theta_1)$ **then**
            $\Theta_e \leftarrow \bar{\Theta} + \beta(\Theta_r - \bar{\Theta})$
            **if** $L(\Theta_e) < L(\Theta_r)$ **then**
                $\Theta_{n+1} \leftarrow \Theta_e$
            **else**
                $\Theta_{n+1} \leftarrow \Theta_r$
            **end if**
        **else if** $L(\Theta_n) \leq L(\Theta_r) < L(\Theta_{n+1})$ **then**
            $\Theta_{oc} \leftarrow \bar{\Theta} + \gamma(\Theta_r - \bar{\Theta})$
            **if** $L(\Theta_{oc}) \leq L(\Theta_r)$ **then**
                $\Theta_{n+1} \leftarrow \Theta_{oc}$
            **end if**
        **else if** $L(\Theta_r) \geq L(\Theta_{n+1})$ **then**
            $\Theta_{ic} \leftarrow \bar{\Theta} - \gamma(\Theta_r - \bar{\Theta})$
            **if** $L(\Theta_{ic}) < L(\Theta_{n+1})$ **then**
                $\Theta_{n+1} \leftarrow \Theta_{ic}$
            **end if**
        **end if**
        **for** $i \leftarrow 1, 2, \cdots, n + 1$ **do**
            $\Theta_i \leftarrow \Theta_1 + \delta(\Theta_i - \Theta_1)$
        **end for**
        Sort all $\Theta_i$ s.t. $L(\Theta_1) < L(\Theta_2) < \cdots < L(\Theta_{n+1})$
    **end for**
    **return** $\Theta \leftarrow \Theta_1$

---

Beta(4.06,...) accounts for 3/8 of the total number of instances, that is, 480000 instances. This part of the instances is divided equally by time windows with densities of 25%, 50%, 75%, and 100%, respectively. Therefore, the instances with time window constraint density of 25% account for $1/4$ of them, which is 120000 instances. The value range [22.17, 39.39] means that for the instances distributed as Beta(4.06,...), the time window half-widths are randomly generated according to the probability density of Beta(4.06,...) only in the range of [22.17, 39.39], and the probability outside the range is zero. In the RC series, "Within Instances, Generated Proportionally" means that for the same instance within 100 customer points, different customer points may have different time window half-widths. Taking RC1 as an example, in the same instance, the time window half-widths of 1/4 of the customer points, that is, 25 customer points, are 5, and the 25 customer points are 60, and the remaining 60 customer points are of distribution Beta(1.94,87.21,8.89,663.77).

A.4 PERFORMANCE OF ALGORITHMS ON SOLOMON BENCHMARK

Table 5 shows the length of routings solved by DRL algorithms and traditional algorithms on Solomon benchmark.

Table 3: The distribution of each instance.

| Instances | Distribution functions | Instances | Distribution functions |
|---|---|---|---|
| C101 | Beta(4.06,5.95,16.05,35.34) | R202 | 75% of R201 |
| C102 | 75% of C101 | R203 | 50% of R201 |
| C103 | 50% of C101 | R204 | 25% of R201 |
| C104 | 25% of C101 | R205 | constant 120 |
| C105 | Beta(3.66,5.33,33.51,67.03) | R206 | 75% of R205 |
| C106 | Gamma(1.52,12.49,43.03) | R207 | 50% of R205 |
| C107 | constant 90 | R208 | 25% of R205 |
| C108 | Beta(3.73,5.23,66.20,133.38) | R209 | Beta(1.30,2.27,44.52,359.15) |
| C109 | constant 180 | R210 | Beta(0.90,1.76,36.50,457.83) |
| C201 | constant 80 | R211 | GE(0.22,222.49,34.74) |
| C202 | 75% of C201 | RC101 | constant 15 |
| C203 | 50% of C201 | RC102 | 75% of RC101 |
| C204 | 25% of C201 | RC103 | 50% of RC101 |
| C205 | constant 160 | RC104 | 25% of RC101 |
| C206 | Beta(3.67,5.20,133.29,266.06) | RC105 | 1/4 constant 5, 1/4 constant 60, |
| C207 | Beta(0.86,1.41,88.50,547.94) | | 1/2 Beta(1.94,87.21,8.89,663.77) |
| C208 | constant 320 | RC106 | constant 30 |
| R101 | constant 5 | RC107 | 1/2 Beta(2.88,8.24,19.28,40.81), |
| R102 | 75% of R101 | | 1/2Beta(12.26,10.26,16.42,78.39) |
| R103 | 50% of R101 | RC108 | Beta(9.90,5.49,-27.18,129.57) |
| R104 | 25% of R101 | RC201 | constant 60 |
| R105 | constant 15 | RC202 | 75% of RC201 |
| R106 | 75% of R105 | RC203 | 50% of RC201 |
| R107 | 50% of R105 | RC204 | 25% of RC201 |
| R108 | 25% of R105 | RC205 | 1/4 constant 30, 1/4 constant 240, |
| R109 | GE(0.23,27.77,4.35) | | Weibull(2.05, 92.65, 31.63) |
| R110 | Beta(1.23,1.82,11.32,79.54) | RC206 | constant 120 |
| R111 | Beta(0.77,1.25,9.50,88.05) | RC207 | Beta(1.30, 2.27, 44.52, 359.15) |
| R112 | GE(0.24,55.60,8.57) | RC208 | GE(0.22, 222.48, 34.73) |
| R201 | GE(0.22,51.24,17.33) | | |

## A.5 FURTHER APPLICATION ON HOMBERGER BENCHMARK

Compared to the Solomon benchmark, which is a commonly used public test set of instances in the CVRPTW field with a general customer size, the Homberger benchmark is another public test set of instances in the CVRPTW field with a larger customer size, ranging from 200 to 1000. We expanded its instances using the method proposed in this paper, as shown in Table 6. Correspondingly, we trained the DRL algorithm using the expanded instances, and compared the resulting model with traditional algorithms on Homberger benchmark. The experimental results are shown in Table 7.

## A.6 VISUALIZATION OF GENERATED INSTANCES

To better provide some visualizations of the generated instances, a comparison of the time window half-width distribution between the original benchmark and the expanded data is presented in Figure 6, which include both frequency histograms and frequency curves. All instances from all series are merged together for statistics. The number of instances in the expanded data used for comparison is the same as the number of instances in the original benchmark. Instances with a constant time window half-width are not included . For instances where the widths of some customer points are constant, those customer points are also not included.

Table 4: The distribution of each series for generating DER-Solomon.

| Series | Distribution functions | Value ranges | Percentage |
|---|---|---|---|
| C 1 | Beta(4.06,5.95,16.05,35.34) | [22.17, 39.39] | 3/8(25%,···,100% ) |
| | Beta(3.66,5.33,33.51,67.03) | [44.49, 78.79] | 1/8 |
| | Gamma(1.52,12.49,43.03) | [20.38, 182.43] | 1/8 |
| | Beta(3.73,5.23,66.20,133.38) | [88.87, 157.46] | 1/8 |
| | constant 90 and 180 | 90,180 | 1/4(90 and 180 ) |
| C 2 | constant 80 | 80 | 1/2(25%,···,100% ) |
| | constant 160 and 320 | 160,320 | 1/4(160 and 320 ) |
| | Beta(3.67,5.20,133.29,266.06) | [177.81, 315.13] | 1/8 |
| | Beta(0.86,1.41,88.50,547.94) | [100.08, 561.94] | 1/8 |
| R 1 | constant 5 | 5 | 1/4(25%,···,100% ) |
| | constant 15 | 15 | 1/4(25%,···,100% ) |
| | GE(0.23,27.77,4.35) | [22.35, 37.11] | 1/8 |
| | Beta(1.23,1.82,11.32,79.54) | [15.38, 77.68] | 1/8 |
| | Beta(0.77,1.25,9.50,88.05) | [10.91, 87.51] | 1/8 |
| | GE(0.24,55.60,8.57) | [44.82, 73.69] | 1/8 |
| R 2 | GE(0.22,51.24,17.33) | [29.68, 88.83] | 3/8(25%,···,100% ) |
| | constant 120 | 120 | 1/4(25%,···,100% ) |
| | Beta(1.30,2.27,44.52,359.15) | [61.63, 322.55] | 1/8 |
| | Beta(0.90,1.76,36.50,457.83) | [45.57, 403.76] | 1/8 |
| | GE(0.22,222.49,34.74) | [179.26,297.74] | 1/8 |
| RC1 | constant 15 | 15 | 1/2(25%,···,100% ) |
| | Within instances, generated proportionally: | | 1/8 |
| |   1/4constant 5, | 5 | |
| |   1/4constant 60, | 60 | |
| |   1/2Beta(1.94,87.21,8.89,663.77) | [22.52, 39.25] | |
| | constant 30 | 30 | 1/8 |
| | Within instances, generated proportionally: | | 1/8 |
| |   1/2Beta(2.88,8.24,19.28,40.81), | [22.52,39.25] | |
| |   1/2Beta(12.26,10.26,16.42,78.39) | [45.65,72.19] | |
| | Beta(9.90,5.49,-27.18,129.57) | [29.53, 79.97] | 1/8 |
| RC2 | constant 60 | 60 | 1/2(25%,···,100% ) |
| | Within Instances, Generated Proportionally: | | 1/8 |
| |   1/4constant 30, | 30 | |
| |   1/4constant 240, | 240 | |
| |   Weibull(2.05, 92.65, 31.63) | [45.15, 140.14] | |
| | constant 120 | 120 | 1/8 |
| | Beta(1.30, 2.27, 44.52, 359.15) | [61.63, 322.55] | 1/8 |
| | GE(0.22, 222.48, 34.73) | [179.26, 297.74] | 1/8 |

Table 5: Performance of DRL algorithms and traditional algorithms on Solomon benchmark.

| Solomon instances | Optimal | DRL | | | Tradition | | | |
|---|---|---|---|---|---|---|---|---|
| | | AM | MDAM | POMO | GA | LKH | OR-Tools | HGS |
| C101 | 827.3 | **828.9** | **828.9** | 829.7 | **827.3** | 828.9 | 917.6 | **827.3** |
| C102 | 827.3 | 942.5 | 958.9 | **898.5** | 858.4 | 828.9 | 1008.6 | **827.3** |
| C103 | 826.3 | 987.4 | 987 | **928.6** | 857.4 | 828.1 | 974.6 | **826.3** |
| C104 | 822.9 | 993 | 959.5 | **875.4** | **822.9** | 824.8 | 1084 | **822.9** |
| C105 | 827.3 | **828.9** | **828.9** | 829.7 | **827.3** | 828.9 | 893.4 | **827.3** |
| C106 | 827.3 | **828.9** | 828.9 | 829.7 | **827.3** | 829.4 | 904.8 | **827.3** |
| C107 | 827.3 | **828.9** | 830.5 | 831.2 | **827.3** | 828.9 | 829.7 | **827.3** |
| C108 | 827.3 | **828.9** | 876.7 | 832.7 | **827.3** | 829.4 | 831.7 | **827.3** |
| C109 | 827.3 | **833.2** | 871.4 | 835.2 | 828.4 | 828.9 | 830.9 | **827.3** |
| C201 | 589.1 | 744.2 | 753.4 | **591.6** | **589.1** | 591.6 | 591.6 | **589.1** |
| C202 | 589.1 | 810.5 | 838.3 | **675.6** | **589.1** | 591.6 | 664.2 | **589.1** |
| C203 | 588.7 | 817.4 | 874.3 | **673.5** | **588.7** | 591.6 | 624.5 | **588.7** |
| C204 | 588.1 | 814.2 | 823.7 | **684.8** | 621.4 | 591.2 | 664.3 | **588.1** |
| C205 | 586.4 | 728.8 | 764.9 | **588.9** | **586.4** | 588.9 | 631.1 | **586.4** |
| C206 | 586 | 730.5 | 760.1 | **588.9** | **586** | 588.5 | 592.3 | **586** |
| C207 | 585.8 | 697.3 | 750.5 | **588.9** | **585.8** | 588.5 | 591.4 | **585.8** |
| C208 | 585.8 | 719.8 | 768.5 | **588.9** | **585.8** | 588.3 | 594.3 | **585.8** |
| R101 | 1637.7 | 1786.4 | 1754.2 | **1695.9** | 1673.6 | 1617.2 | **1583.8** | 1637.7 |
| R102 | 1466.6 | 1618.6 | 1625.2 | **1538** | 1472.6 | 1458.9 | **1432.6** | 1466.6 |
| R103 | 1208.7 | 1315.4 | 1382.2 | **1303.2** | 1220.6 | 1219.4 | 1219.5 | **1208.7** |
| R104 | 971.5 | 1073.5 | 1095.2 | **1060.7** | 989.2 | 1003.9 | 1097.6 | **971.5** |
| R105 | 1355.3 | 1493.7 | 1476.4 | **1447.2** | 1369.3 | 1360.6 | 1411.5 | **1355.3** |
| R106 | 1234.6 | 1344.6 | 1393.5 | **1305.9** | 1242.7 | 1245 | 1303.4 | **1234.6** |
| R107 | 1064.6 | 1180 | 1180.1 | **1170.2** | 1080.5 | 1084.5 | 1124.7 | **1064.6** |
| R108 | 932.1 | 1041.7 | 1035 | **992.6** | 970.0 | 952.5 | 1016 | **932.1** |
| R109 | 1146.9 | 1270.9 | 1299 | **1231.1** | 1168.9 | 1151.3 | 1238.7 | **1146.9** |
| R110 | 1068 | 1185.6 | 1230 | **1158.5** | 1128.9 | 1099 | 1196.9 | **1068** |
| R111 | 1048.7 | 1162.7 | 1212.8 | **1150.1** | 1058.8 | 1082.8 | 1132.5 | **1048.7** |
| R112 | 948.6 | **997.1** | 1079.5 | 1026.2 | 963.8 | 971 | 1073.1 | **948.6** |
| R201 | 1143.2 | 1272.1 | 1294.1 | **1237.9** | 1163.0 | **1143.2** | 1215.6 | 1143.2 |
| R202 | 1029.6 | 1199.9 | 1214 | **1144.6** | 1036.6 | 1031.7 | 1139.3 | **1029.6** |
| R203 | 870.8 | 998 | 1061.5 | **964.2** | 913.3 | 877.5 | 955.3 | **870.8** |
| R204 | 731.3 | 824.4 | 881.4 | **763.4** | 765.3 | 735.9 | 802.3 | **731.3** |
| R205 | 949.8 | 1045.1 | 1074.4 | **1007.8** | 981.4 | 955 | 1003.4 | **949.8** |
| R206 | 875.9 | 1006.7 | 1047.5 | **967.4** | 915.6 | 888.4 | 1019.9 | **875.9** |
| R207 | 794 | 891.2 | 982.2 | **877.5** | 825.4 | 816.9 | 917.4 | **794** |
| R208 | 701 | 769 | 778 | **740.4** | 754.9 | 707.5 | 795.2 | **701** |
| R209 | 854.8 | 951.9 | 1001.2 | **943.3** | 872.3 | 866.3 | 955.9 | **854.8** |
| R210 | 900.5 | 1045.3 | 1080.7 | **984.7** | 923.8 | 916.1 | 956.8 | **900.5** |
| R211 | 746.7 | 837.6 | 845.4 | **806.3** | 782.7 | 763.2 | 824.3 | **746.7** |
| RC101 | 1619.8 | 1754.1 | 1822.4 | **1715.1** | 1638.3 | 1650.4 | 1666.8 | **1619.8** |
| RC102 | 1457.4 | 1592.3 | 1670.2 | **1578.7** | 1492.9 | 1482.2 | 1527.6 | **1458** |
| RC103 | 1258 | 1441.9 | 1508.8 | **1323.3** | 1318.5 | 1282.9 | 1364.1 | **1258** |
| RC104 | 1132 | 1240.7 | 1224.8 | **1201.9** | 1201.9 | 1136.8 | 1313.9 | **1132.3** |
| RC105 | 1513.7 | 1652 | 1699 | **1621.2** | 1586.0 | 1518 | 1519.6 | **1513.7** |
| RC106 | 1372.7 | 1519.6 | 1580.5 | **1467.6** | 1391.0 | 1404.2 | 1461 | **1372.7** |
| RC107 | 1207.8 | 1306.1 | 1422.7 | **1292** | 1217.6 | **1206.6** | 1380.4 | 1207.8 |
| RC108 | 1114.2 | 1220 | 1276.3 | **1196.2** | 1141.5 | 1135.4 | 1282.2 | **1114.2** |
| RC201 | 1261.8 | 1359 | 1366.9 | **1343.5** | 1272.8 | 1274.5 | 1368.9 | **1261.8** |
| RC202 | 1092.3 | 1212.5 | 1249.9 | **1169** | 1114.7 | 1104.3 | 1157.5 | **1092.3** |
| RC203 | 923.7 | 1071.6 | 1083.8 | **996.1** | 961.3 | 942.6 | 1043.1 | **923.7** |
| RC204 | 783.5 | 925.5 | 920 | **873.2** | 803.2 | 787.5 | 850 | **783.5** |
| RC205 | 1154 | 1279.8 | 1270.4 | **1218.6** | 1159.3 | 1163.1 | 1322.7 | **1154** |
| RC206 | 1051.1 | 1164.8 | 1208 | **1140.6** | 1078.7 | 1053.6 | 1179.4 | **1051.1** |
| RC207 | 962.9 | 1083.3 | 1136.9 | **1067.3** | 966.6 | 968.9 | 1018 | **962.9** |
| RC208 | 776.1 | **838.6** | 916.8 | 866.9 | 791.6 | 790.7 | 912.3 | **776.1** |

Table 6: The distribution of each series for the generation of Homberger extension instances.

| Series | Distribution functions | Value ranges | Percentage |
|---|---|---|---|
| C 1 | Weibull(1.2911, 29.761, 4.3503) | [21.46, 38.06] | 2/5(25%,···,100%) |
| | logNorm(0.025096, -345.58, 405.81) | [43.83, 77.34] | 1/10 |
| | Beta(2.698, 5.0564, -8.1061, 252.09) | [19.04, 152.02] | 1/10 |
| | constant 90 | 90 | 1/10 |
| | logGamma(823.03, -3770.5, 579.59) | [86.47, 152.95] | 1/10 |
| | constant 180 | 180 | 1/10 |
| | logNorm(0.017227, -2387.4, 2625.6) | [164.85, 313.67] | 1/10 |
| C 2 | constant 80 | 80 | 2/5(25%,···,100%) |
| | constant 160 | 160 | 1/10 |
| | t(80.569, 248.69, 44.064) | [175.36, 322.01] | 1/10 |
| | logGamma(1500.3, -39813, 5486.5) | [76.62, 542.66] | 1/10 |
| | constant 320 | 320 | 1/10 |
| | Beta(4.7071, 7.5232, -129.09, 1410.8) | [118.92, 740.22] | 1/10 |
| | constant 440 | 440 | 1/10 |
| R 1 | constant 5 | 5 | 2/5(25%,···,100%) |
| | constant 15 | 15 | 2/5(25%,···,101%) |
| | Beta(19.967, 21.871, -31.535, 129.45) | [14.05, 46.60] | 1/10 |
| | Beta(43.76, 63.152, -68.136, 314.14) | [36.28, 85.22] | 1/10 |
| R 2 | Beta(17.91, 23.268, -50.954, 255.94) | [28.65, 93.00] | 2/5(25%,···,100%) |
| | constant 120 | 120 | 2/5(25%,···,101%) |
| | Beta(3.5593, 6.148, -29.968, 587.98) | [52.94, 337.54] | 1/10 |
| | Gamma(7620.6, -2395.3, 0.34582) | [190.60, 289.91] | 1/10 |
| RC1 | constant 15 | 15 | 2/5(25%,···,100%) |
| | Beta(3.7284, 7.0289, -3.7358, 100.58) | [9.90, 55.75] | 1/10 |
| | constant 30 | 30 | 1/10 |
| | GE(0.25447, 39.379, 14.724) | [20.74, 70.07] | 1/10 |
| | t(8580.8, 59.812, 14.947) | [35.22, 84.40] | 1/10 |
| | constant 60 | 60 | 1/10 |
| | constant 75 | 75 | 1/10 |
| RC2 | constant 60 | 60 | 2/5(25%,···,100%) |
| | Beta(2.0915, 5.6153, -5.7556, 524.29) | [26.75, 283.33] | 1/10 |
| | constant 120 | 120 | 1/10 |
| | t(21377, 184.78, 85.286) | [44.50, 325.07] | 1/10 |
| | Norm(238.07, 88.745) | [92.10, 384.05] | 1/10 |
| | constant 240 | 240 | 1/10 |
| | constant 300 | 300 | 1/10 |

We found that the distribution of instances of different customer sizes in Homberger is very close. Therefore, we have integrated the data of all customer point sizes in the same series to backward derive the distribution of time window half-widths of this series. Function Weibull, Beta and GE are given by formulae (1)-(3), other are given as follows:

logNorm$(x; s, loc, scale) = \frac{1}{sy\sqrt{2\pi}}\exp(-\frac{log^2(y)}{2s^2})$.

logGamma$(x; c, loc, scale) = \frac{\exp(cy - \exp(y))}{\Gamma(c)}$, where $\Gamma$ is the gamma function as formulae (4).

Gamma$(x; a, loc, scale) = \frac{y^{a-1}e^{-y}}{\Gamma(a)}$.

t$(x; v, loc, scale) = \frac{\Gamma((v+1)/2)}{\sqrt{\pi v}\Gamma(v/2)}(1 + x^2/v)^{-(v+1)/2}$.

Norm$(x; loc, scale) = \frac{exp(-y^2/2)}{\sqrt{2\pi}}$.

Where $y = \frac{x-loc}{scale}$.

Table 7: Performance of a DRL algorithm and traditional algorithms on part Homberger benchmark.

| Homberger instances | Optimal | DRL POMO | Tradition GA | LKH | OR-Tools | HGS |
|---|---|---|---|---|---|---|
| C1_2_1 | 2698.6 | 2706.6003 | 2713.2 | 2704.6 | 4296.2 | **2698.6** |
| C1_2_2 | 2694.3 | 2794.6006 | 2738.4 | 2917.9 | 4245.5 | **2694.3** |
| C1_2_3 | 2675.8 | 2819.4006 | 2740.7 | 2707.3 | 4217 | **2675.8** |
| C1_2_4 | 2625.6 | 4150.2012 | 2723.1 | 2643.3 | 4145.2 | **2625.6** |
| C1_2_5 | 2694.9 | 2712.2002 | **2694.9** | 2702 | 4197.4 | **2694.9** |
| C1_2_6 | 2694.9 | 2707.8 | **2694.9** | 2701 | 4185 | **2694.9** |
| C1_2_7 | 2694.9 | 2703.6003 | **2694.9** | 2701 | 4244.5 | **2694.9** |
| C1_2_8 | 2684 | 2722.2998 | 2687.7 | 2775.5 | 4505.7 | **2684** |
| C1_2_9 | 2639.6 | 2722.2998 | 2650.3 | 2687.8 | 4281.7 | **2639.6** |
| C1_2_10 | 2624.7 | 2703.5994 | 2670.6 | 2643.5 | 4302.5 | **2624.7** |
| C1_4_1 | 7138.8 | 7177.1055 | 7633.8 | 7152 | 9723.9 | **7138.8** |
| C1_4_2 | 7113.3 | 7548.7051 | 7165.4 | 7687.4 | 9672.6 | **7113.3** |
| C1_4_3 | 6929.9 | 7504.103 | 7204.1 | 7065.6 | 9804.2 | **6930.3** |
| C1_4_4 | 6777.7 | 7279.2017 | 7161.5 | 6803.1 | 9397.8 | **6799.4** |
| C1_4_5 | 7138.8 | 7155.4067 | 7337.3 | 7152 | 9483.9 | **7138.8** |
| C1_4_6 | 7140.1 | 7161.3062 | 7167.4 | 7153.4 | 9855.9 | **7140.1** |
| C1_4_7 | 7136.2 | 7172.4038 | 7298.5 | 7417.9 | 10120.3 | **7136.2** |
| C1_4_8 | 7083 | 7179.5054 | 7172.1 | 7365.2 | 10253.3 | **7084.9** |
| C1_4_9 | 6927.8 | 7170.3047 | 7270.5 | 7068.5 | 9820.1 | **6942.5** |
| C1_4_10 | 6825.4 | 7130.7041 | 7245.5 | 6863.6 | 9756.2 | **6826.7** |
| C1_6_1 | 14076.6 | 15025.4014 | 17103.9 | 14095.5 | 17705 | **14076.6** |
| C1_6_2 | 13948.3 | 15126.9971 | 14907.5 | 14163.1 | 17247.2 | **13953** |
| C1_6_3 | 13756.5 | 14865.7939 | 14739.2 | **13777.6** | 17286.9 | 13834.1 |
| C1_6_4 | 13538.6 | 14441.3945 | 14654.6 | 13607.9 | 17559.4 | **13604.4** |
| C1_6_5 | 14066.8 | 15145.1045 | 14532.7 | 14085.6 | 17561.7 | **14067.7** |
| C1_6_6 | 14070.9 | 15768.3008 | 14360.7 | infeasible | 18030 | **14070.9** |
| C1_6_7 | 14066.8 | 14857.4043 | 14556.9 | infeasible | 18184 | **14067.7** |
| C1_6_8 | 13991.2 | 14340.5 | 14578.9 | 14815.3 | 18110.6 | **13999.8** |
| C1_6_9 | 13664.5 | 14302.7031 | 14523.3 | **13719.7** | 17765.1 | 13720.7 |
| C1_6_10 | 13617.5 | 14405.3994 | 14859.2 | **13664.6** | 18258.5 | 13666.4 |
| C1_10_1 | 42444.8 | 47295.2031 | | | 43785.5 | **42444.8** |
| C1_10_2 | 41337.8 | 46488.2148 | | | 43926 | **41723.5** |
| C1_10_3 | 40064.4 | 45299.0117 | | | 41619.2 | **40792.8** |
| C1_10_4 | 39434.1 | 43110.4922 | | | **39675.3** | 40147.4 |
| C1_10_5 | 42434.8 | 47927.582 | | | 45555.5 | **42434.8** |
| C1_10_6 | 42437 | 50163.0938 | | | 47451.2 | **42437** |
| C1_10_7 | 42420.4 | 46748.2969 | | | 45130 | **42422.4** |
| C1_10_8 | 41652.1 | 44636.8086 | | | 47819.4 | **42251.4** |
| C1_10_9 | 40288.4 | 44079.2031 | | | 44383 | **41243.9** |
| C1_10_10 | 39816.8 | 44048.0273 | | | 44279.1 | **40644.8** |

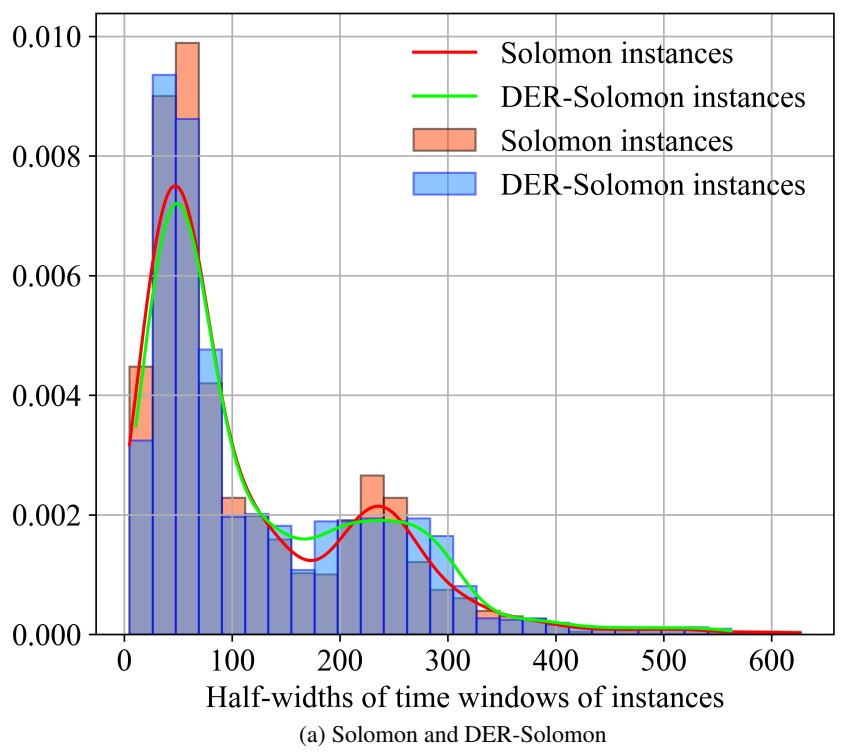

(a) Solomon and DER-Solomon

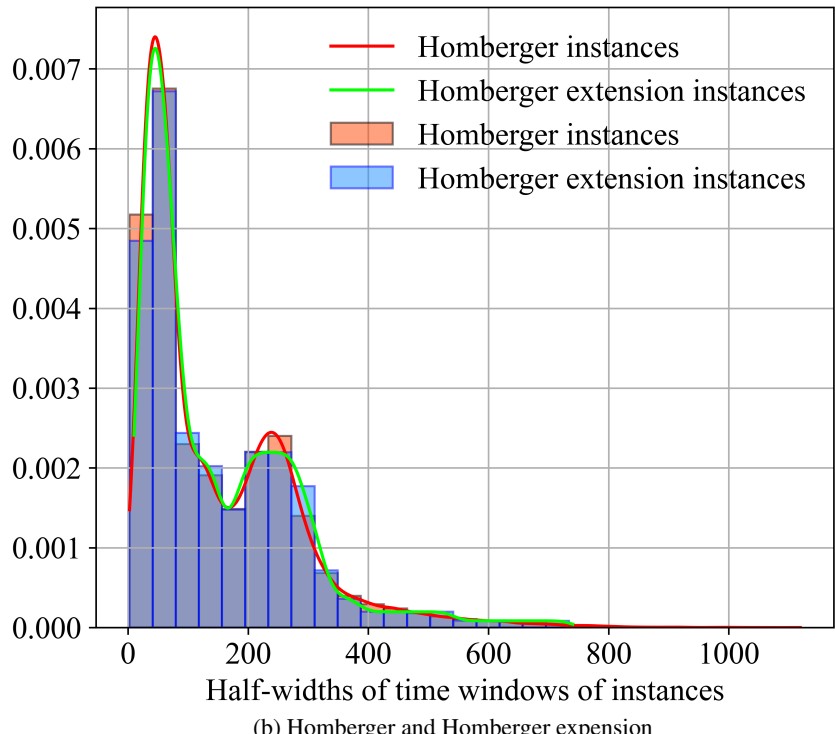

(b) Homberger and Homberger expension

Figure 6: Frequency comparison between origin benchmark and expanded data

