# OpenReview forum: "DER-Solomon: A Large Number of CVRPTW Instances Generated Based on the Solomon Benchmark Distribution"
_ICLR.cc/2024/Conference — Submitted to ICLR 2024_

### Official Review · Reviewer_D9sp · 2023-10-18

**Soundness:** 2 fair
**Presentation:** 2 fair
**Contribution:** 3 good
**Rating:** 3
**Confidence:** 3

**Summary:**

This is a dataset and benchmarking paper. The author proposes DER-Solomon, an expanded CVRPTW dataset, by approximating the original Solomon dataset with backward derivation. Experiments are conducted on traditional and DL-based algorithms to demonstrate its merits.

**Strengths:**

* Compared with Solomon-like datasets, the proposed method could better approximate the original Solomon dataset.
* The proposed dataset has values for further research in CVRPTW.

**Weaknesses:**

* The practicality is not clear.
  * Is there any evidence to demonstrate that, with only 56 instances, the original Solomon benchmark is complex enough to test *all aspects* of (traditional or DL-based) algorithms? If a company trains a DL-based model on DER-Solomon, could we guarantee its reliability in practice?
  * Besides Solomon, is the proposed method generalizable to approximate other datasets?
* It seems only the distribution of the time window is approximated. However, other attributes, such as the customer location, may significantly affect the learned policy as well. Have you considered the variations of all attributes in DER-Solomon?
* For the traditional algorithms, it would be better to add HGS. For the DL-based method, some recent studies [1, 2] demonstrate superior performance on CVRPTW, it would be better to benchmark them as well.
* The provided link to the source code cannot be opened.
* The writing and presentation of this paper should be improved:
  * All figures should use PDF format. The current version is blurry when zooming in.
  * Better to provide some visualizations of the generated instances.
  * The best result (e.g., in all tables) should be in bold for a better view.

[1] Learning to delegate for large-scale vehicle routing. In NeurIPS 2021.
[2] RBG: Hierarchically solving large-scale routing problems in logistic systems via reinforcement learning. In KDD 2022.

----

**Overall,** this paper only focuses on *one dataset of a single problem (i.e., CVRPTW)*, and therefore the contribution may not be enough for ICLR. Currently, I lean towards rejection, and I may adjust the evaluation after reading other reviews and the author's rebuttal.

**Questions:**

* Will you release the source code and datasets?

---

> ### Author Response · Authors · 2023-11-20
> **Response to Reviewer D9sp**
>
> Thank you for your helpful comments and great efforts in helping improve our work. The responses to your comments are listed as follows.
>
> > Q1. The practicality is not clear: Is there any evidence to demonstrate that, with only 56 instances, the original Solomon benchmark is complex enough to test all aspects of (traditional or DL-based) algorithms? If a company trains a DL-based model on DER-Solomon, could we guarantee its reliability in practice?
>
> A1. Thank you for your insightful comments. It’s challenging to directly apply the model trained on the Solomon benchmark to real-world problems. We need to further collect real-world data and use our method to expand the collected data as training instances. Since most traditional algorithms demonstrate their performance on the Solomon benchmark, our work in this paper mainly constructs a dataset for training so that DRL algorithms can also show their inherent performance on the Solomon benchmark. This allows for a fair comparison between traditional algorithms and DRL algorithms, accelerating algorithm research in this field.
> > Q2. Is the proposed method generalizable to approximate other datasets?
>
> A2. Thank you for your helpful comments. We have included additional expanded instances, validated through a comparison experiment on the Homberger benchmark, in the appendix of the updated manuscript. In the final version of the paper, we will add approximations for most of the public test benchmark in the VRP field. Due to time limits, experiments on them will be gradually released in the open-source code.
>
> > Q3. It seems only the distribution of the time window is approximated. However, other attributes, such as the customer location, may significantly affect the learned policy as well. Have you considered the variations of all attributes in DER-Solomon?
>
> A3. Thank you for your thoughtful comments. We did take into account the point you raised, but we concluded that for a DRL method, it is not necessary to account for variations in customer coordinates between instances of the same series in the Solomon benchmark, as they remain fixed.
>
> > Q4. For the traditional algorithms, it would be better to add HGS. For the DL-based method, some recent studies [1, 2] demonstrate superior performance on CVRPTW, it would be better to benchmark them as well.
>
> A4. Thank you for your thoughtful comments. Both papers used HGS, thus, we believe it should be a very useful solver. However, we noticed that HGS only has CVRP code implementation, and we are unsure whether it can be applied to CVRPTW. Article [1] is a great work, and the key is that there is a source code implementation. The README file is written in great detail, and the author have put a lot of effort into it. Thank you very much for recommending this article. We are very willing to use this algorithm in our experiment. Article [2] is also a great work, but we do not know whether it has source code published. If no source code is published and the author has not published the experimental results of the algorithm on the Solomon benchmark, it is challenging to conduct an experiment on it.
>
> > Q5. The provided link to the source code cannot be opened.
>
> A5. Thank you for your insightful comments. Please compare the link you entered into the browser with the link in the pdf file. Because the underscore “_” may be lost during the pasting process, we made a reminder at the first source code reference on page 7 of the article.

---

> > ### Author Response · Authors · 2023-11-20
> > **Response to Reviewer D9sp**
> >
> > > Q6. The writing and presentation of this paper should be improved: All figures should use PDF format. The current version is blurry when zooming in.
> >
> > A6. Thank you for your helpful comments. In update manuscript, Figures 1 and 3 have been converted to PDF format in the updated manuscript. Figure 2 may be clear enough. The original Figure 4 has been deleted (as it duplicates the content of Table 1, and there is a 9-page limit). The new Figures 4 and 5 both have used high-resolution images (300dpi).
> >
> > > Q7. The writing and presentation of this paper should be improved: Better to provide some visualizations of the generated instances.
> >
> > A7. Thank you for your helpful comments. To better provide some visualizations of the generated instances, we have added a distribution function diagram in the appendix. Please let us know if there are any remaining questions!
> > > Q8. The best result (e.g., in all tables) should be in bold for a better view.
> >
> > A8. Thank you for your helpful comments. In the updated manuscript, in Table 1, we have bolded the maximum Gap between DER-Solomon and Solomon. In Table 2, we have marked the shortest computation time for the DRL algorithm and the traditional algorithm, respectively. In Appendix Table 5, we have marked the optimal values for the DRL algorithm and the traditional algorithm, respectively.
> >
> >
> > > Q9. Overall, this paper only focuses on one dataset of a single problem (i.e., CVRPTW), and therefore the contribution may not be enough for ICLR. Currently, I lean towards rejection, and I may adjust the evaluation after reading other reviews and the author's rebuttal.
> >
> > A9. Thank you for your encouraging comments and great efforts in helping improve our work. Indeed, the entire VRP field needs such work. If you look at the articles on DRL methods in the VRP field, you will find that they basically (to the best of our knowledge) test the performance of the algorithm with the author's custom dataset. In contrast, non-learning-based traditional methods (those algorithms that do not need to use other data for training) all use public test data to demonstrate algorithm performance. This has led to a lack of comparison between the two types of algorithms and a gap between the two fields.
> > If we could expand the entire VRP field dataset, it would be very meaningful.
> >
> > However, many validation experiments need to be done, such as proving the similarity between the expanded data and the original data through the similarity of the solution results of the traditional algorithm on the original data and the expanded data. For example, use the expanded data to train the DRL method, and then compare it with the traditional method on the public test data, and show that using this method can indeed make the DRL method have a good performance on the public test data, so as to attract future DRL method authors to use this method to show the performance of their algorithm on the public test data, because I believe that if this method cannot make the DRL method perform well on the public test data, I am afraid that no author is willing to put their carefully developed algorithm in a not objective and fair scenario to deliberately show poor performance.
> >
> > The workload of so many experiments on various VRP benchmark data is very huge, and we will proceed with the following work step by step:
> >
> > First, expand the Homberger benchmark dataset and conduct corresponding validation experiments on the expanded dataset.
> >
> > Then, reverse deduce the public test dataset of various VRP variants, expand it, and publish the expanded dataset.
> >
> > Finally, conduct experimental verification on the expanded dataset.
> >
> > > Q10. Will you release the source code and datasets?
> >
> > A10. We will share our source code and datasets in the final version of our paper.

---

> > ### Comment · Reviewer_D9sp · 2023-11-21
> > **Reply to Rebuttal**
> >
> > Hi authors, thanks for your detailed responses. Please see the comments below:
> >
> > > We concluded that for a DRL method, it is not necessary to account for variations in customer coordinates between instances of the same series in the Solomon benchmark, as they remain fixed.
> >
> > It seems that the practicality is not the first concern of this paper. Since in real-world cases, customer coordinates or demands may vary a lot. I still think considering more variations in different dimensions may improve the significance of the proposed dataset.
> >
> > > However, we noticed that HGS only has CVRP code implementation, and we are unsure whether it can be applied to CVRPTW.
> >
> > See https://github.com/ortec/euro-neurips-vrp-2022-quickstart/tree/main/baselines/hgs_vrptw

---

> > > ### Author Response · Authors · 2023-11-21
> > > **Response to Reviewer D9sp**
> > >
> > > Thank you for your precious time, thoughtful comments, and great efforts to help improve our work. The responses to your comments are listed as follows.
> > >
> > > Q1. It seems that the practicality is not the first concern of this paper. Since in real-world cases, customer coordinates or demands may vary a lot. I still think considering more variations in different dimensions may improve the significance of the proposed dataset.
> > >
> > >  A1. Thank you for your thoughtful comments. The deep learning correlation algorithm can adapt to the problem of random customer points very well. But, the traditional algorithm needs to be modified in order to adapt. In this work, we focused on a fair comparison between the two types of approaches and made the most of the available publicly performance metrics, so we didn't make too many changes to the original Solomon.
> > >
> > > In order to test the performance of each algorithm more comprehensively, in future work, we will take into account the changes of each variable, and even the correlation between variables, and the variable may present a multi-peak distribution. Our approach works well to achieve these goals, and it's just a matter of workload.
> > >
> > >
> > > Q2. See https://github.com/ortec/euro-neurips-vrp-2022-quickstart/tree/main/baselines/hgs_vrptw
> > >
> > > A2. Thank you for your helpful comments. According to the code link provided by the reviewer, HGS can be well applied to CVRPTW. We have conducted relevant research on HGS and are conducting supplementary experiments. The experimental results will be added to the final version of the paper.

---

> > > > ### Comment · Reviewer_D9sp · 2023-11-22
> > > > **Rebuttal Ack**
> > > >
> > > > Thanks for considering my comments. Unfortunately, based on the current version, I cannot recommend acceptance. I hope the authors could further improve the presentation of the paper. Building a project page may help as well. Then, submit it to another venue (e.g., the benchmark track of NeurIPS).

---

> > > > > ### Author Response · Authors · 2023-11-22
> > > > > **Response to Reviewer D9sp**
> > > > >
> > > > > Thank you for your precious time, helpful comments, and great efforts to help improve our work. The responses to your comments are listed as follows.
> > > > >
> > > > > Q. Thanks for considering my comments. Unfortunately, based on the current version, I cannot recommend acceptance. I hope the authors could further improve the presentation of the paper. Building a project page may help as well. Then, submit it to another venue (e.g., the benchmark track of NeurIPS).
> > > > >
> > > > > A. Thank you for your thoughtful and helpful comments. According to the code link you provided, HGS can be well applied to CVRPTW after working all night. We have conducted relevant research on HGS and have conducted supplementary experiments in Table 5 and Table 7. The experimental results are added to the updated version of the paper.
> > > > >
> > > > > If you have any suggestions for improvement, please share them. I appreciate your help.

---

### Official Review · Reviewer_c63s · 2023-10-28

**Soundness:** 3 good
**Presentation:** 2 fair
**Contribution:** 2 fair
**Rating:** 3
**Confidence:** 4

**Summary:**

The paper studies to use deep reinforcement learning to solve the classic optimization problem of capacitated vehicle routing problem with time windows. To make DRL possible, the paper proposes to create a training dataset that is based on the Solomon dataset that is small but comprehensive to test algorithms for CVRPTW problem. The paper creates the new DER-Solomon dataset by first estimating the probability distributions of the essential parameters of the Solomon dataset and then generating new problem instances by using the estimated distributions. The paper then trains a DRL model on the DER-Solomon dataset and shows comparable performance on the testing instances compared to optimization-based approaches.

**Strengths:**

1. The paper proposes a approach to effectively enlarge the training data for DRL on a specific classic optimization problem given a small but comprehensive instance set.

2. The paper shows that the learned DRL with the enriched dataset could achieve comparable performance compared to classic optimization methods.

**Weaknesses:**

1. The paper lacks background descriptions of the CVRP and CVRPTW problems and also probably some more detailed introductions on the existing traditional algorithm approaches. Given there is plenty of space for the paper, such background information could be very beneficial to the audience. VRP may be well-known in the community, but the variants are probably not.

2. A lot of details are not presented in the paper. E.g., given the estimated distributions of the parameters, how are the new instances sampled? Are all parameters considered independently? Why generate 1280k instances? Moreover, there are no given details on how the DRL model is trained using DER-Solomon.

3. The technical contribution is limited. It mainly estimates the distributions of a given small dataset to generate instances to form a larger dataset. It does not compare the proposed sampling method to some more basic methods. For example, what if we just add Gaussian noises to parameters in Solomon or use some uniform sampling to generate the instances?

4. Curretly I think parameters of the dataset are assumed to be independently samples (correct me if that is wrong). Is there a reason to make such an assumption? Would it be beneficial to consider a more complex distribution of the parameters?

5. The studied problem has a relative restricted scope. Could such techniques explored in the paper get applied to solving other classic optimization problems as well? Or what special properties of the Solomon dataset makes the approach most effective?


Minor:
1. Page 3:  "its frequency histogram is shown in Figure 2(a)" but there is no index of (a) or (b) in Figure 2.
2. X-axis of Figure 5 is not labeled.

**Questions:**

Please check the weaknesses part.

---

> ### Author Response · Authors · 2023-11-20
> **Response to Reviewer c63s**
>
> Thank you for your insightful comments and great efforts in helping improve our work. The responses to your comments are listed as follows.
>
> > Q1. The paper lacks background descriptions of the CVRP and CVRPTW problems and also probably some more detailed introductions on the existing traditional algorithm approaches. Given there is plenty of space for the paper, such background information could be very beneficial to the audience. VRP may be well-known in the community, but the variants are probably not.
>
> A1. Thank you for your helpful comments. To make the background descriptions clearer, we have added more background information on the CVRP and CVRPTW, as well as detailed introductions to existing traditional algorithm approaches in the updated manuscript.
>
> > Q2. A lot of details are not presented in the paper. E.g., given the estimated distributions of the parameters, how are the new instances sampled? Are all parameters considered independently? Why generate 1280k instances? Moreover, there are no given details on how the DRL model is trained using DER-Solomon.
>
> A2.  Thank you for your insightful comments. We sincerely apologize for any lack of detail in certain areas. Given the complexity of the subject matter, it can be challenging to cover every aspect in depth. To address this, we plan to release the source code in the final version of the paper. This will allow for a more comprehensive understanding as many details can be directly gleaned from the code.
>
> - New instances are generated by random sampling from the estimated distributions.
> - All parameters are considered independently. However, in Solomon's public test instance and another public test instance of CVRPTW, Homberger changes between instances, except for the time window variable, while other variables remain fixed. So, we also kept other variables fixed in the new instance, and only one variable changed. Whether other variables are independent of each other is irrelevant.
> - The reason for generating 1280k instances is that most of the DRL algorithms solving VRP after 2019 are based on the transformer network and attention mechanism. The first author [1] to use this method set the training dataset to 1280k instances in his open-source code, so most subsequent researchers have followed suit.
> - We didn’t mentioned in this paper how the DRL model is trained using DER-Solomon because the cited DRL algorithms, AM, MDAM, POMO, and MARDAM, all have source code released, and their articles also detailed their network structure and training methods. As for the training process, in short, the coordinates, demand, and time window data of DER-Solomon are input into the DRL network, and then the DRL network will plan a path that meets the time window and load constraints for each instance. The negative value of the average path length is used as the reward value, and the DRL network is trained using the gradient optimization algorithm.
>
>
> [1] Kool, W, Van Hoof H, and Welling M. "Attention, Learn to Solve Routing Problems!." International Conference on Learning Representations. 2018.
>
>
> > Q3. The technical contribution is limited. It mainly estimates the distributions of a given small dataset to generate instances to form a larger dataset. It does not compare the proposed sampling method to some more basic methods. For example, what if we just add Gaussian noises to parameters in Solomon or use some uniform sampling to generate the instances.
>
> A3. Thank you for your thoughtful comments. In fact, not only public test instances, but our original starting point was to consider the problems that may be encountered in actual situations when the available dataset is small. Imagine there is a logistics company, or a company that provides decision system solutions to logistics companies. In order to seize the first-mover advantage in the market, it’s crucial to act proactively when there are indications of a developing business scenario. This involves implementing an intelligent logistics and transportation planning system even before the scenario fully matures. At this time, the instances that can be collected in the business scenario are very few, and it may not even represent the characteristics of all future instances, but it needs to speculate on all future instances. Therefore, we assumed 11 forms of popular distribution functions and selected the one with the best goodness of fit. The 11 forms of distribution functions are the crystallization of human statistical technology over the past hundreds of years, not just a summary of the current few instances. This cannot be achieved by adding Gaussian noise and random sampling. The instances generated by Gaussian noise can only represent the characteristics of the current few instances, to a certain extent, they are just replicas of the current instances.

---

> > ### Author Response · Authors · 2023-11-20
> > **Response to Reviewer c63s**
> >
> > > Q4. Curretly I think parameters of the dataset are assumed to be independently samples (correct me if that is wrong). Is there a reason to make such an assumption? Would it be beneficial to consider a more complex distribution of the parameters
> >
> > A4. Thank you for your suggestion, we indeed did not consider this. In the Solomon public test instances, except for the time window variable is changing, other variables (e.g. coordinates, demand, etc.) are fixed, so we didn't think about whether the parameters were independently sampled. But considering that we will extend this method to the entire VRP field next, we can try to analyze the relationship between parameters, and then use the joint probability distribution to constrain non-independent parameters.
> >
> > > Q5. The studied problem has a relative restricted scope. Could such techniques explored in the paper get applied to solving other classic optimization problems as well? Or what special properties of the Solomon dataset makes the approach most effective
> >
> > A5. Thank you for your insightful comments. Our method can be applied to optimization problems in various fields, not just the Solomon dataset, even when limited by a small number of public test instances. We have included additional expanded instances, validated through a comparison experiment on the Homberger benchmark, in the appendix of the updated manuscript. In the final version of the paper, we will add approximations for most of the public test data in the VRP field. Due to time limits, experiments on them will be gradually released in the open-source code.
> >
> > > Q6. Minor
> >
> > A6. Thank you for your insightful comments. The details of the figures have been updated in the updated manuscript.

---

### Official Review · Reviewer_Cpep · 2023-10-31

**Soundness:** 2 fair
**Presentation:** 3 good
**Contribution:** 3 good
**Rating:** 6
**Confidence:** 3

**Summary:**

To deal with the limited scale of the well-known Solomon benchmark of the capacitated vehicle routing problem with time windows (CVRPTW) for learning-based approaches, this paper proposes a large set of new instances with a similar distribution to the Solomon benchmark, called DER-Solomon benchmark.

**Strengths:**

To deal with the limited scale of the well-known Solomon benchmark of the capacitated vehicle routing problem with time windows (CVRPTW) for learning-based approaches, this paper proposes a large set of new instances with a similar distribution to the Solomon benchmark, called DER-Solomon benchmark.

**Weaknesses:**

Besides the Solomon benchmark, the Gehring & Homberger benchmark is also a famous one of CVRPTW. It is better to further apply the proposed data generation method to the Gehring & Homberger benchmark.

**Questions:**

1. How many instances are included in the DER-Solomon benchmark?
2. How many DER-Solomon instances are used to train the learning-based method?
3. What and how many original Solomon instances are used to train the learning-based method?
4. How to calculate the std gaps reported in Table 1? They seem to be unequal to the relative gaps between the std values of two benchmarks.

---

> ### Author Response · Authors · 2023-11-20
> **Response to Reviewer Cpep**
>
> Thank you for your encouraging comments and great efforts in helping improve our work. The responses to your comments are listed as follows.
>
> > Q1. Besides the Solomon benchmark, the Gehring & Homberger benchmark is also a famous one of CVRPTW. It is better to further apply the proposed data generation method to the Gehring & Homberger benchmark.
>
> A1.
> Thank you for your insightful suggestion. We have recorded the expanded dataset of the Gehring & Homberger benchmark in the appendix of the updated manuscript, along with some of our experimental results. The experimental verification process is time-consuming, given that each of the six series requires individual training. Furthermore, each series encompasses five different customer point scales, which also necessitate separate training. This results in a total of 30 training models, and that is just for one Deep Reinforcement Learning (DRL) method.
>
> If we consider all three DRL methods mentioned in the article (i.e. AM, POMO, and MDAM), we would need to train a staggering 90 models. However, we will continue to update the remaining experiments and those extended to other optimization problems periodically on the source code, much like the author of AM, who has diligently maintained his source code for over two years.
>
>
> > Q2. How many instances are included in the DER-Solomon benchmark.
>
> A2. As briefly mentioned in Appendix A.3, we used 1,280,000 DER-Solomon instances as the training set. In addition, during testing, 1024 instances were used as the test set in Tables 1 and 2. However, it's important to note that the DER-Solomon benchmark is not a fixed set of instances. It is more of an instance generation method that provides a distribution function of instances, generating a large number of training instances through random sampling to help the DRL method test its performance on Solomon.
>
> > Q3. How many DER-Solomon instances are used to train the learning-based method.
>
> A3. As briefly mentioned in Appendix A.3, we used 1,280,000 DER-Solomon instances as the training set.
>
> > Q4. What and how many original Solomon instances are used to train the learning-based method.
>
> A4.  We opted not to use the original Solomon instances as part of the training set, given that they are ultimately designated as the test set. Instead, all training data exclusively comprises DER-Solomon instances.
>
> > Q5. How to calculate the std gaps reported in Table 1? They seem to be unequal to the relative gaps between the std values of two benchmarks
>
> A5. We always set the denominator to be the mean. The calculation method of the gap is:
>
> std gap: abs (std on Solomon – std on DER-Solomon ) / mean on Solomon.
>
> mean gap: abs (mean on Solomon – mean on DER-Solomon ) / mean on Solomon.
>
> And we have explained it in Table 1 in the updated manuscript.

---

> > ### Comment · Reviewer_Cpep · 2023-11-22
> >
> > Thanks for your responses. I keep my borderline rating due to three points as follows.
> >
> > 1. The contribution is relatively limited. The proposed method should be applied to more problems besides CVRPTW.
> > 2. Other data generation method to better train the DRL method should be compared.
> > 3. When the dataset is small in some realistic situations, do we really need to separately train a DRL method? Instead, a conventional OR method might be a better choice.
> >
> > I encourage the authors to further improve the paper in the next version.

---

> > > ### Author Response · Authors · 2023-11-22
> > > **Response to Reviewer Cpep**
> > >
> > > Thank you for your precious time, thoughtful comments, and great efforts to help improve our work. The responses to your comments are listed as follows.
> > >
> > > Q1. The contribution is relatively limited. The proposed method should be applied to more problems besides CVRPTW.
> > >
> > >  A1. Thank you for your thoughtful comments. We proposed a new data generation method by backward deriving that was able to scale not only in the SOMOLON benchmark but also in other combinatory-optimized traditional benchmark datasets such as the Homberger benchmark. Other variants of VRP benchmark experimental results are added to the final version of the paper.
> > >
> > > Q2. Other data generation method to better train the DRL method should be compared.
> > >
> > > A2. Thank you for your helpful comments. We will compare our method with those that generate data using Gaussian noise and random sampling. If you have other recommended data generation methods, we look forward to your suggestions.
> > >
> > > Q3.	When the dataset is small in some realistic situations, do we really need to separately train a DRL method? Instead, a conventional OR method might be a better choice.
> > >
> > > A3. Thank you for your thoughtful comments. In some practical situations, the traditional OR method may be better for small datasets, while the DRL method may excel in larger datasets with high efficiency.
> > >
> > > We welcome any further questions or discussions if any point is unclear. Looking forward to your reply, thank you very much!

---

### Official Review · Reviewer_mp8y · 2023-11-02

**Soundness:** 3 good
**Presentation:** 2 fair
**Contribution:** 2 fair
**Rating:** 3
**Confidence:** 3

**Summary:**

This work scales up the Solomon benchmark with backward deriving.  Distribution consistency has been verified between the generated dataset and the original one.

**Strengths:**

1. Effect solution to scale up the Solomon benchmark.
2. Code is publicly available.

**Weaknesses:**

1. While effective, the contribution of this work is quite limited. I suggest authors consider applying this algorithm to scale more benchmarks.
2. The comparison with neural solvers is missed in Table  1.

**Questions:**

The figures are not well presented. Also hard to find the relevant descriptions. This paper is not ready to be published.

---

> ### Author Response · Authors · 2023-11-20
> **Response to Reviewer mp8y**
>
> Thank you for your thoughtful comments and great efforts to help improve our work. The responses to your comments are listed as follows.
>
> > Q1. The comparison with neural solvers is missed in Table 1.
>
> A1. The purpose of Table 1 is to compare the differences between DER-Solomon and Solomon, not the differences between different algorithms. To clarify the manuscript, we have added vertical lines between algorithms and have made annotations at the bottom of Table 1. In addition, the comparison between algorithms are recorded in appendix Table 5.
>
>
> > Q2. The figures are not well presented. Also hard to find the relevant descriptions. This paper is not ready to be published.
>
> A2. Thank you for your thoughtful comments. To make the figures present clear, we have described more content in legends of the figures and have made some essential explanations in the titles of the figures. We have utilized the PDF format for some Figures while rendering others in higher resolution for improved visual quality.

---

> ### Comment · Reviewer_mp8y · 2023-11-23
>
> Thanks for your effort and timely reply, I will keep my score. In addition, I strongly encourage authors to consider applying this algorithm to scale more benchmarks.

---

### Author Response · Authors · 2023-11-20
**Common Response**

We sincerely appreciate all reviewers' time and efforts in reviewing our paper. We first highlight the significance of this work.

• In this paper, we proposed a new data generation method by backward deriving that was able to scale not only in the SOMOLON benchmark but also in other combinatory-optimized traditional benchmark datasets such as the Homberger benchmark.

• The goal of this paper is to conduct a fair comparison between traditional algorithms and DRL algorithms in order to expedite the research process. While traditional algorithms are tested on public benchmarks to demonstrate their capabilities, DRL algorithms rely on datasets they define themselves. To address this difference, we have developed a dedicated dataset for training DRL algorithms, which enables them to showcase their performance on public benchmarks.

• In practical applications, our method addresses the challenge of data scarcity in emerging logistics scenarios. By generating data that mirrors real-world characteristics, we enable DRL models to be effectively trained, providing logistics companies with a competitive edge in new markets.

Following constructive feedback from reviewers, we have updated our manuscript, which mainly includes further application to other benchmarks, the updated Figures and Tables, and additional background descriptions. Specifically, we revise our paper in the following aspects.

•  [Further application] In this paper, we proposed a versatile data generation method that can be applied to various benchmarks in optimization problems. We have included additional expanded instances, validated through a comparison experiment on the Homberger benchmark, in the appendix of the updated manuscript. In the final version of our paper, we plan to propose expanded instances for some public test benchmarks in other fields of Vehicle Routing Problem (VRP). Additionally, we are committed to further expanding our method to other optimization problems and conducting additional experiments. These will be gradually released in our open-source code, providing valuable resources for the research community and stimulating further advancements in the field.

• [Tables and Figures] We have incorporated concise annotations at the bottom of the Tables, and the title position of the majority of the Figures and the legends are enhanced with additional information for better understanding, enabling readers to grasp the intended meaning more effectively. Furthermore, we have utilized the PDF format for some Figures while rendering others in higher resolution for improved visual quality.

• [Background descriptions] The introduction adds more background information on the CVRP and CVRPTW problems.

Please kindly see the updated manuscript and reevaluate our work. We provide detailed pointwise responses to each reviewer below and hope they can address all reviewers' questions and concerns.

We welcome any further questions or discussions if any point needs to be clarified. We are looking forward to your reply!

---

### Meta-Review · Area_Chair_gLXJ · 2023-12-08

**Metareview:**

Majority of the reviewers have raised significant issues with the paper, and I agree with many of the concerns raised. I would encourage the authors to carefully look through the reviews and try to address them in the next submission!

**Justification For Why Not Higher Score:**

The paper has significant issues that I do not think can be addressed in a rebuttal. The paper would benefit from one more round of reviewing and I would encourage the authors to address the issues and resubmit to a different venue.

**Justification For Why Not Lower Score:**

N/A

---

### Decision · Program_Chairs · 2024-01-16

Reject